# Analysis of the Technological Convergence in Smart Textiles

**Qian Xu [1,2]** , **Yabin Yu [1,2,\*]** and **Xiao Yu [1]**

1   School of Economics and Management, China Jiliang University, Hangzhou 310018, China
2   School of Economics and Management, Zhejiang Sci-Tech University, Hangzhou 310018, China
*   Correspondence: yyb311300@163.com

**Abstract:** Convergence between emerging technologies and traditional industries has become a crucial strategy for enhancing a technology's competitiveness. Technical convergence (TC) for smart textiles aims to reveal the convergence of emerging technologies with textile technologies, including the field, structure, and critical technologies of the TC. For the empirical analysis, the technology life cycle (TLC) and network analysis method are utilized to observe the TC of 15,125 patent data for textiles from the Derwent Patent Database. The results indicate the following: (1) after 2021, the TC of smart textiles matured, with the number of patents reaching a peak in 2030. (2) Emerging technologies and textile technologies are inextricably linked. In addition to textile technologies, the primary technical fields involved in smart textiles are electronic engineering, tools design, chemical engineering, and mechanical engineering. Electronic engineering is the most common of these fields, accounting for 29.11%. (3) From a structural perspective, the density, breadth, and depth of the TC continues to expand. (4) Measurement, computer technology, and audio technology will be always essential to the TC, whereas electrical machinery, instrumentation, energy technology, other specialized technologies, and chemical engineering have tremendous growth potential. The findings above have substantial implications for the phenomenon of the TCs that have emerged in emerging technology and traditional industry fields. They can also aid the government in formulating policies that promote the transformation and growth of related industries.

**Keywords:** technological convergence; smart textile; traditional industry; technology life cycle; network analysis

## 1. Introduction

In recent years, numerous innovative, disruptive technologies in information, biology, energy, materials, and manufacturing have rapidly penetrated industry and spread in recent years, accelerating the growth of emerging industries and having a profound effect on traditional industries [1]. Technological convergence (TC) drives emerging technologies such as cloud computing, big data, and blockchain [2]. It provides businesses with a significant competitive advantage [3]. In addition, it offers businesses the potential to overcome technological bottlenecks in product development and capacity expansion and is anticipated to lead the next generation of technological innovation [4].

The convergence of emerging technologies and traditional industries has emerged as a new mean for nations to enhance their competitiveness [5]. However, most current research focuses on the TCs of high-tech fields and industries. For example, the convergence of robot technology [1,6], the Internet of Things (IoT), and information and communication technology (ICT) have been discussed at the technical field level [7,8]. In addition, the biological information industry [9] and the ICT industry [10] have been studied extensively at the industrial level.

In the meantime, the TCs of traditional industries have been disregarded. Consequently, this study focuses on smart textiles, a representative example of emerging technologies converging with the textile industry, which historically characterized as a labor-intensive, less innovative industry [11].

Smart textiles are woven, spun, or braided using smart fibers or other smart materials [11–13]. As the key application object of the industrial internet, as well as the excellent carrier of intelligent services, smart textiles have generated considerable interest in the industry. The United States Joint Market Research (AMR) has revealed that the global smart textile market will grow from USD 943 million in 2015 to USD 5.369 billion in 2022.

This study aims to investigate the fusion between textile technology and emerging digital technology. Specifically, what is the technological life cycle of smart textiles? What technical fields are involved in smart textiles' TC? What is the structure of the TC? What are the primary technologies? For the purpose of achieving these objectives, the life cycle of smart textiles is classified into three development stages. The composition, structure, and key technologies of the TC in each stage are then determined by utilizing the social network method. The significance of our research is: (1) to lay a theoretical foundation for the future technology integration of emerging technologies and traditional industries. (2) Network analysis indicators can be employed to establish the relationship between technical fields and determine the rules of the TC while providing academic references. (3) By analyzing the technological cooperation trend of emerging technologies in the textile industry, new technological opportunities can be found, and future research and development trends can be predicted, which will help the government in formulating industrial policies.

The remainder of this study is organized as follows: Section 2 provides the literature review and hypotheses; Section 3 discusses the methodology; Section 4 contains the empirical analysis results; and Section 5 covers the discussion and the conclusion.

## 2. Literature and Review

### 2.1. Technology Life Cycle

The evolution of technology is analogous to the evolution of life, which goes through stages of birth, growth, maturity, decline, and death. Therefore, the evolution of technology will progress through various stages and exhibit various characteristics. According to the technology life cycle (TLC) theory, technology development can be divided into four stages [14]: (1) during the germination stage, few technological innovations occur, and most those that do are fundamental, and the technical direction is uncertain; (2) throughout the growth phase, new technologies continue to spread throughout the industry while technology is becoming more appealing, and more innovative entities invest in research and development [11]; (3) research and development technology has reached maturity, but the number of patent applicants remains relatively constant while the rate of technological innovation slows as a result of the market's restriction; and (4) technology is aging during the recession as profitability declines and companies exit the market, resulting in a decline in technological innovations [9].

The S-shaped growth curve is a frequently used method for evaluating the TLC [15]. Chen et al. (2010) [16] compared the life cycles and development potential of two emerging technologies, hydrogen energy and fuel cells, and forecast their future development. Daim et al. (2006) [17] analyzed three emerging technologies: fuel cells, food safety, and optical storage. Liu and Wang (2010) [15] used an S-curve and a logistic model to forecast the development trend of biped robot walking technology in Japan. Finally, Huang et al. (2017) [18] applied an S-curve and a logistic model to analyze the development trajectory of 3D printing technology; they believed that the technology began to sprout in 1985, entered the growth stage in 2005, and would enter the mature stage in 2016.

### 2.2. Technology Convergence

A TC refers to the convergence and penetration of multiple technology fields [19]. A TC is typically measured through network analysis. Therefore, studying a TC via social networking is feasible and practicable [20]. Furthermore, network analysis can aid in understanding the state of a TC at each stage and the intricate interaction between technologies during the TC process [5]. Network analysis indicators include both local and global indicators. Local indicators include node strength and link coefficients. Meanwhile,

global indicators include the network's size, density, and average degree. Several studies on TCs have examined the type, scale, heterogeneity, and other local indicator characteristics, while others have examined the network's global indicators [21].

Existing research on TCs is primarily concerned with establishing the phenomenon's mode, level, and evolution path. Choi et al. (2015) [22] examined the mode of TC diffusion using Korean patent data and a logistic model and found that the forms of interdepartmental diffusion were diverse. Lee (2007) [6] examined the mode and process of technology fusion through the lens of intelligent robot technology. There are two distinct types of technological fusion: one is fusion in which key technologies are combined to form a variety of new technologies and the other is subtyping, wherein two different technologies combine to form completely new technology. Lee et al. (2015) [23] predicted technology fusion patterns using ternary patent data via association rules and link prediction methods, and they used topic models to discover and predict emerging areas of technology fusion.

A TC's structure and mode of action have been the subject of increasing research. Kim et al. (2014) [24] investigated the structure of a TC through the lens of printing electronics and proposed core technologies at various stages. Lee et al. (2007) [6] predicted the future trend of TCs using association rules and link prediction methods and discovered emerging areas of TCs using topic models. Choi et al. (2015) [22] recognized the heterogeneity of TCs and classified it as cross-departmental and cross-field in nature; the findings indicated that cross-departmental diffusion takes on a more diverse form.

## 3. Data and Methodology

### 3.1. Data

Patent data is widely regarded as the most trustworthy source of knowledge (Lee et al., 2015) [23]. Typically, a patent is comprised of numerous technical classification numbers. Therefore, the technical information contained is relatively comprehensive, serving as a barometer of smart textile technology innovation and indicating the direction of technological change. The patent data in this study is from the Derwent Patent Database (DII), which is widely used for technological opportunity discovery research. The search strategy for this study's smart textiles is summarized in Table 1. First, this study created the search formula TS = ("textile" or "textiles" or "cloth" or "clothing") to conduct a keyword search for the term "textile." The search returned 584,990 records, with the search number being #1. Then, for the "smart" keyword, the search formula TS = ("smart" or "intelligent" or "wearable") was constructed, and the subject search was conducted. A total of 854,539 records were obtained, with the search number being #2. Finally, the two search results above (i.e., #1 and #2) were combined, yielding 15,125 search results. The search period was 2000–2019, with the search date set to 21 November 2020.

**Table 1.** Retrieval structure.

| Number | Result | Formula |
|---|---|---|
| #3 | 15,125 | #1 and #2 |
| #2 | 854,539 | TS = ("smart" or "Intelligent" or "wearable") |
| #1 | 584,990 | TS = ("textile" or "textiles" or "cloth" or "clothing") |

### 3.2. Methodology

#### 3.2.1. S-Curve Model

Richard Foster (1986) [25], an American scholar, proposed the S-curve model in his book to exhibit the relationship between three periods: growth, maturity, and recession. Kim (2003) [14] proposed that TLC assessments be made using patent data. By establishing a visual graph with time as the horizontal axis and the cumulative number of patents as the vertical axis, this technological development trend illustrates evolution similar to the English letter "S". The changes in the curve correspond to various stages of the TLC, as illustrated in Figure 1.

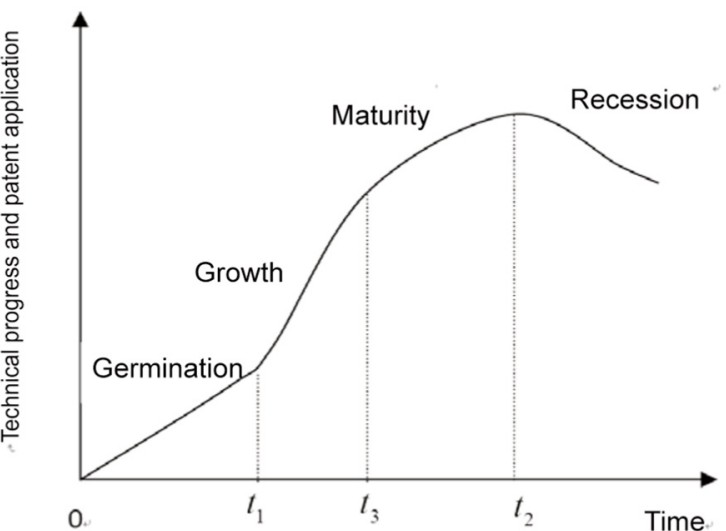

**Figure 1.** S-curve.

### 3.2.2. Identification of the TC

Since a TC is predicated on the technological frontier, the division of technology fields is critical. The World Intellectual Property Organization (WIPO) established the International Patent Classification (IPC), widely used in many countries worldwide. The co-occurrence analysis method identifies the fusion of various technologies based on multiple technical categories in a single patent, and each patent can determine its technical category based on IPC information. Moreover, this method can identify the technical category directly and accurately [21,23]. Hence, this study analyzed the TC using the patent co-occurrence approach.

The IPC system is based on product and technology classification. Therefore, it does not allow for the analysis of a TC within an industry. WIPO's ISI-OST-INPI classification system can accurately map IPC categories to industrial technologies. By mapping between IPC subcategories and technology sectors, it is possible to identify cross-industry and cross-sectoral TCs.

### 3.2.3. Construction of the TC Network

A TC denotes combining knowledge or innovations from various technological fields to create a dominant invention [24]. These fields will eventually be connected via the social network theory to form a TC network. Hence, existing technological elements coexist with those of new technologies. In this case, a new invention incorporates multiple existing technology fields, and the resulting phenomenon of the technology co-occurrence exhibits evident network characteristics, forming a TC network.

In this study, BibExcel was used to calculate the frequency and co-occurrence of IPC numbers and construct an IPC co-occurrence matrix. The matrix was then used to construct the IPC co-occurrence network, and finally, the IPC number was mapped to the domain number in the ISI-OST-INPI classification system, yielding the technical field co-occurrence network, as illustrated in Figure 2.

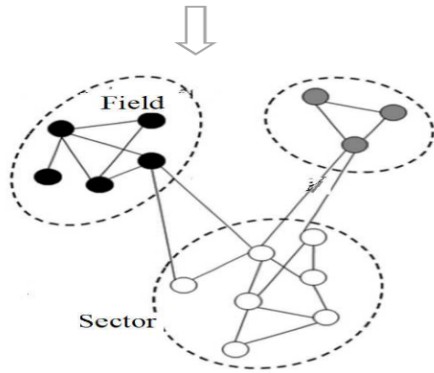

**Figure 2.** The construction of a TC network.

3.2.4. Network Analysis

(1) Number of network nodes

The scale of a TC is defined by the number of nodes in the TC network or the number of technical fields represented in the TC network of smart textiles. The more nodes, the more technical fields involved in the TC, and the larger the network scale.

$$\text{Number of network nodes} = \frac{1}{2}\sum_{i}^{n}\sum_{j}^{n} l_{ij}, \ i \neq j \text{ and } l_{ij} = \begin{cases} 1, & m_{ij} > 0 \\ 0, & else \end{cases}$$

where $m_{ij}$ represents the frequency of the co-occurrence of technical field i and technical field j, while n represents the total number of network nodes.

(2) Density of the TC network

The density of the TC indicates the degree of network aggregation, whereas the density of the co-occurring networks in the technical field denotes the degree of proximity between the technical fields represented by the TC. The value range is [0, 1], and the calculation formula is the number of connections in the technical field's co-occurrence network minus the maximum possible connection ratio. Thus, the number and density of links, where n denotes the total number of nodes, can be expressed as follows:

$$\text{Density} = \frac{1}{2}\frac{\sum_{i}^{n}\sum_{j}^{n} l_{ij}}{{}_{n}C_{2}}, \ i \neq j \text{ and } l_{ij} = \begin{cases} 1, m_{ij} > 0 \\ 0, else \end{cases}$$

where $m_{ij}$ represents the total number of patents where patent i and patent j are fused, and $l_{ij}$ indicates if patent i and patent j are fused.

(3) Average degree of the TC network

The average degree refers to the average degree of all nodes in a network, which reflects the width of the TC. The formula is as follows:

$$\text{Average degree} = \frac{\sum_{i}^{n} l_{i,j}}{2n}, (i \neq j \text{ and } l_{ij} = \begin{cases} 1, m_{ij} > 0 \\ 0, else \end{cases})$$

where $m_{ij}$ represents the total number of patents where patent i and patent j are fused, and $l_{ij}$ indicates if patent i and patent j are fused.

(4)   Weighted average degree

The weighted average degree reflects the depth of crossover or fusion between different technologies in the TC network. By measuring the weighted average degree of the textile technology fusion network in each time period, we can grasp the change in the textile technology fusion intensity, as follows:

$$\text{Weighted average degree} = \sum_{i,j=1}^{n} w_{i,j} / n$$

where $w_{i,j}$ represents the weight of the i-th edge.

(5)   Eigenvector center

The eigenvector center (EC) determines the significance of a single technical field node in the textile technology fusion network. When a node has more neighbors who are also significant, the node is considered significant. When a node's EC value increases, it gains importance in the network, and so EC reflects the key technology in the technology convergence. It is calculated iteratively. The first iteration is obtained by multiplying the adjacency matrix A by the vector of the point degree centrality. The subsequent iterations are the product of the adjacency matrix A and the result of the previous iteration until the convergence point. At this point, the eigenvector's elements represent the EC of each point. The calculation method is as follows:

$$\text{EC(i)} = \lambda \sum_{j=1}^{n} A_{ij} X_j \ (\lambda \text{ is a constant and n is an integer})$$

where EC(i) represents the eigenvector centrality of the i-th node, Aij denotes the i-th row and the j-th column element in matrix A, and Xj is the j-th element in the eigenvector.

## 4. Results

### 4.1. Technology Life Cycle of the TC

Figure 3 is based on the technology life cycle method, which uses a logical model to simulate the development stage of smart textile technology. From 2000 to 2006, there were few patent applications for smart textiles technology, only a handful of companies were conducting research and development, and the application scope of smart textiles technology was limited. For example, Plug and Wear, which sells conductive materials for knitting and sewing, was founded in 2000, and the Georgia Tech Motherboard shirt was printed in 2003.

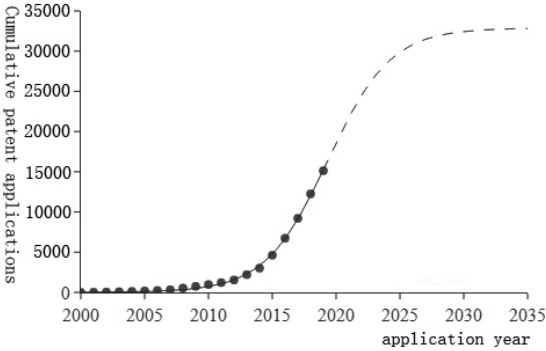

**Figure 3.** Technology life cycle of smart textiles.

From 2007 to 2013, the research and development of smart textile technology entered a period of rapid development, and the range of its applications expanded. In 2009, Forster Rohner introduced the Climate Dress, utilizing its innovative embedded technologies. The MICA Fiber department began experimenting with conductive thread and electronics in 2011, resulting in the Midi Puppet Glove. In 2013, Machine released the Midi Controller Jacket on Kickstarter.

Smart textiles entered a period of rapid growth from 2014 to 2020, and many nations have implemented policies to promote the development of smart textile technology. For instance, "Made in China 2025" clearly defines the intelligent transformation and upgrading of the manufacturing industry, which is crucial in promoting the research, development, and application of intelligent textiles. In addition, the number of patent applications for smart textiles has increased significantly, indicating that smart textile technology has rapidly expanded. In April 2016, collaboration between the DoD and M.I.T was worth USD 302 million and the first-ever U.S. Commerce Department smart-fabric gathering took place. Google's first product, which was slated for release in 2017, was an intelligent jacket, co-developed with Levi's, that featured conductive sensory fibers woven into the sleeve for touch-sensitive mobile phone device control.

Since 2021, the growth of intelligent textile technology has slowed considerably, indicating that it reached its full potential. After 2030, this technology will reach its zenith, intelligent textiles will become increasingly popular, the technical level will be higher, the space for technological advancement will be gradually reduced, and the economy will enter a recession. After a new technology paradigm is generated, a new growth cycle based on the S-curve will commence.

### 4.2. Areas Covered in the TC

Smart textiles result from the penetration and cross-integration of the textile and information technology tracks. In order to understand the fusion structure of smart textiles technology in greater depth, this paper first maps the IPC classification number of 15,125 smart textiles patents to one or more categories based on the standard industrial classification system and then uses the occurrence of a smart textiles technology subcategory to demonstrate the main technical fields and percentage of the TC for smart textiles.

Table 2 outlines the technical classification of smart textiles' top ten integrated patents based on the standard industry and the international patent classification standard. Smart textiles primarily combine the following technological fields: motor, instrument, energy technology, audio-visual technology, telecommunications technology, digital communication technology, computer technology, control technology, medical technology, surface technology, textile technology, and other consumer goods technology. Specifically, 29.1% of the technology comes from electronic engineering, while 13.81% comes from consumer goods-related technology in other fields. Since the industry classification standard places clothing technology within this category, clothing technology is primarily included here. In addition, 13.49% comes from the tool sector, of which 8.83% belongs to medical technology, 4.66% to the surface coating technology of the majority of chemicals, and 7.18% to the textile machinery technology in the mechanical engineering sector.

Table 2 demonstrates that smart textiles are a technology resulting from integrating textiles and information technologies. These include ten major technical fields such as motor, instrument, energy, audio-visual, telecommunications, textile, and medical technology. Apart from information and textile technologies, medical technology is a significant field, consistent with the multiple applications of smart textiles in this industry in recent years, such as smart patient clothing that monitors vital signs such as heart rate, blood pressure, and pulse. This technology can also be integrated into patient pillowcases or bed sheets. The spread of infection is another prominent challenge hospitals face. With antibacterial coatings, smart textiles can solve this issue. This technology could make patient care more efficient and automatic, bringing revolutionary changes to this field.

**Table 2.** Top 10 technology fields of smart textiles.

| Sector | Field | Patent Count | Proportion | IPC |
|---|---|---|---|---|
| Electrical Engineering (29.11%) | 1: Electrical machinery, apparatusand energy | 1066 | 6.29% | H01B, H01R, H01M, H05B, F21V, H01H, H01G, H01C, H01F, H02J, F21L, F21Y, H02N, H02G, F21S, H02K, F21W, H01K, H01J, H02M, H02P, H02H, H02S, H02B, H05F |
| | 2: Audio-visual technology | 737 | 4.35% | H05K, G09F, G09G, H04N, H04R, G11B, H04S |
| | 3: Telecommunications | 739 | 4.36% | H04B, H04M, H04Q, H01Q, H04K, H01P, H04J, G08C, H04H |
| | 4: Digital communication | 684 | 4.04% | H04L, H04W |
| | 6: Computer technology | 1703 | 10.06% | G06F, G06K, G11C, G06T, G06G, G10L, G06N, G06M, G06J |
| Instruments (13.49%) | 12: Control | 788 | 4.65% | G08B, G07C, G07F, G05B, G07B, G07D, G08G, G07G, G09B, G09C, G05D, G05F |
| | 13: Medical technology | 1496 | 8.83% | A61B, A61F, A61N, A61L, A61H, A61M, A61G, G16H, A61J, A61C, A61D |
| Chemistry (4.66%) | 21: Surface technology and coating | 789 | 4.66% | B32B, B05D, C23C, B05C, C25D, C25B, C30B |
| Mechanical Engineering (7.18%) | 28: Textile and paper machines | 1216 | 7.18% | D03D, D02G, D06M, D04B, D01F, A43D, D04H, B41M, A41H, B41J, B41L, D06Q, D03C, D04C, D01D, D01H, D01G, D05B, D06H, D03J, D21H, B41F, D02J, D05C, D06P, C14B, D06G, D01C, B31D, B41C, D21C, A46D, D02H, D01B, D21F |
| Other Fields (13.81%) | 34: Other consumer goods | 2339 | 13.81% | A41D, A44B, A45C, A45F, A41B, B42D, A44C, A42B, B43K, F25D, A43B, A46B, G10K, A41C, A43C, A42C, B44C, A62B, D06N, A45D, D06F, D07B, A41F, A24C, B44F, B68G, G10D, G10G, B44D, A41G, D04D, A45B, B43L, G10H, B68C, B42F |

### 4.3. Structure of the TC

Figures 4–6 depict each stage's technology fusion network diagram based on industrial technology classification standards, with different industrial technology fields as nodes and the fusion relationship between technologies as edges to reveal the status of the TC in smart textiles. 1, 2, 3, 4 and 5 in the figure respectively represent sector of Electrical Engineerin, Instruments, Chemistry, Mechanical Engineering and Other Field.

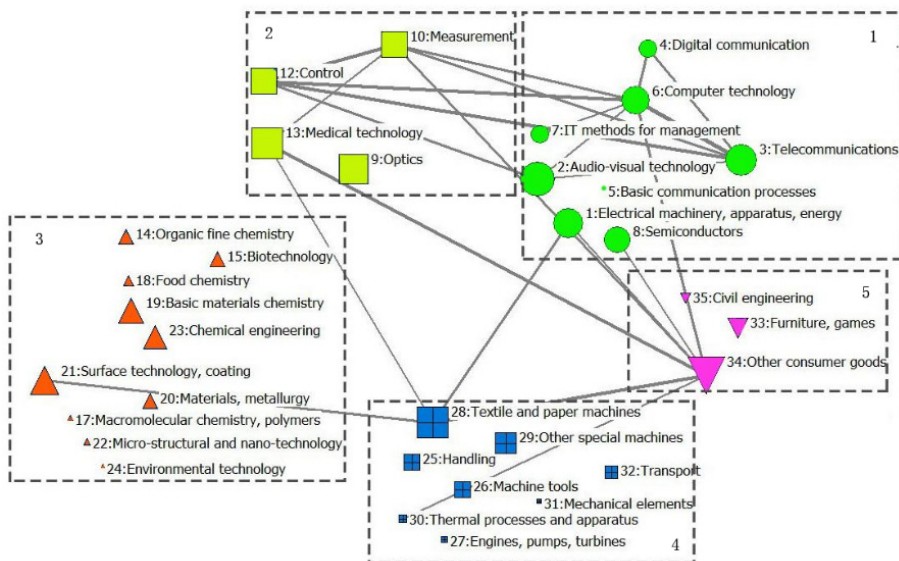

**Figure 4.** TC network for the period 2000–2006.

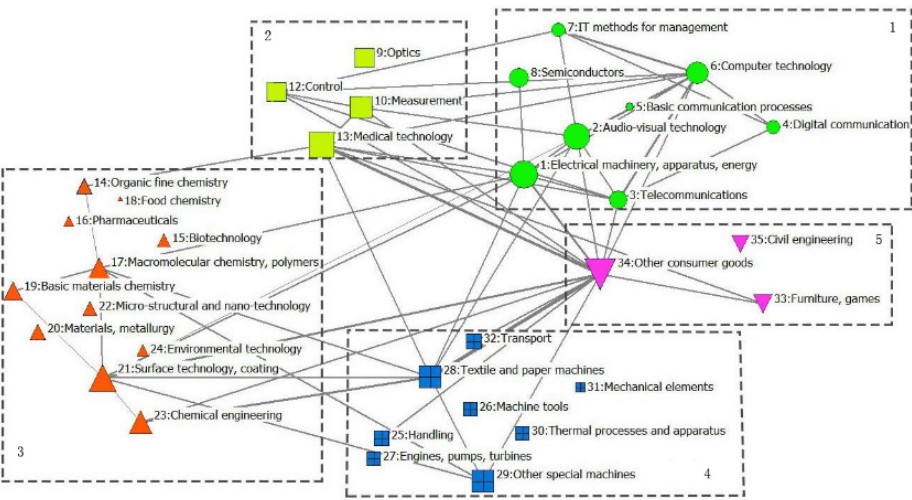

**Figure 5.** TC network for the period 2007–2013.

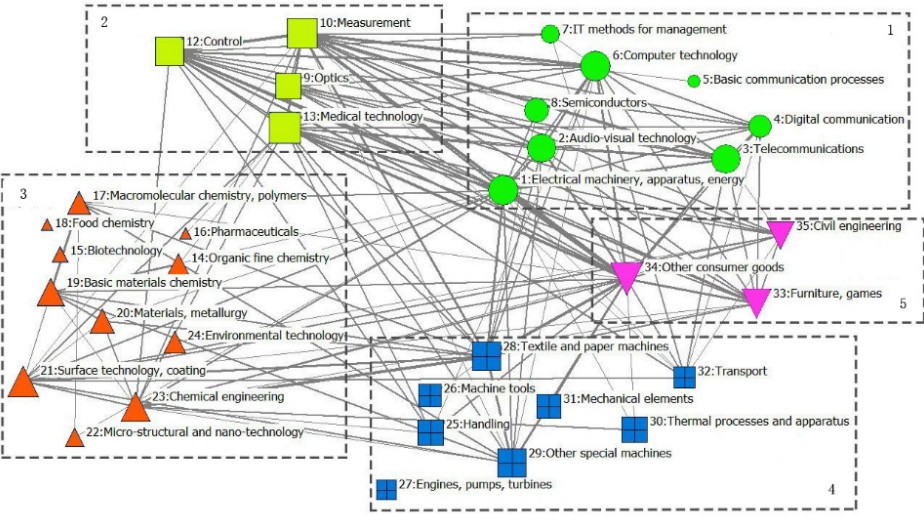

**Figure 6.** TC network for the period 2014–2019.

In Phase I, as depicted in Figure 4, the number of links in the network is small and the degree of TC is low. At this time, the majority of the TC falls under the first category (electronic engineering), the second category (tools), the fourth category (textile and paper machinery), and the fifth category (other consumer goods). In addition, Table 3 reveals that the Phase I indicators are relatively low compared to those of Phases II and III, regardless of the number of network connections, network density, average degree, or weighted average degree. The number of connections is 171, the density is 0.287, and the average degree and weighted average degree of technology fusion are 9.771 and 36,914, respectively. This indicates that the breadth and depth of technology integration in the first stage are limited, as are the number of technical fields involved and the degree f connection between them.

**Table 3.** The indicators of the TC for smart textiles.

| Stage | 2000–2006 | 2007–2013 | 2014–2020 |
|---|---|---|---|
| Node numbers | 171 | 267 | 408 |
| Density | 0.287 | 0.449 | 0.686 |
| Average degree | 9.771 | 15.257 | 23.314 |
| Weighted average degree | 36.914 | 64.514 | 244.286 |

The number of network connections in Phase II has increased significantly, as has the overall technology convergence scale and the number of connections between technology fields. At this stage, the vast majority of technical fields are represented in the TC. Compared to the first stage, the technology of the third category (chemistry) has advanced significantly, and the chemical industry is gaining importance. According to Table 3, the density, breadth, and depth of the TC have significantly increased since the first stage. In Phase II, the network connections increased from 171 to 267. The network density is 0.449, significantly higher than the initial density of 0.287, but the network connectivity has not yet reached 50%. The average degree and weighted average degree of the network are 15.257 and 64.514, respectively, indicating that the number of technical fields involved in this stage has increased significantly, as has the level of integration between technologies.

In Phase III, as depicted in Figure 6, the number of network links and the TC between departments in the network have increased. During the first two phases, technologies with few or no converged nodes were added to the network, such as sectors 9 (optical), 32 (transportation), and 33 (furniture, games). This reflects the evolutionary characteristics of smart textile technologies, on the one hand, and the significance of these technical fields, on the other. As a result, the scale of the technology convergence network is expanding, as are the network's complexity and level of integration, and the integration is strengthening overall. As shown in Table 3, the network density at this stage is 0.686, the network connectivity is high, and the network as a whole is relatively dense. The average degree and weighted average degree reached 23.314 and 244.86, respectively, indicating that the breadth and depth have been further expanded and that the types of technical fields involved in the integration and the proximity between the technical fields have been significantly enhanced.

### 4.4. Key Technology of the TC

When a node has more neighbor nodes, and the neighbor nodes are more important in the TC network, the node can play a significant role in the network and is a key technology. Thus, this study analyzes the network of the three stages above using EC; the specific calculation results are presented in Table 4.

**Table 4.** EC of the top 10 technologies at the three stages.

| 2000–2006 Technology | EC | 2007–2013 Technology | EC | 2014–2019 Technology | EC |
|---|---|---|---|---|---|
| 3: Telecommunications | 1 | 34: Other consumer goods | 1 | 34: Other consumer goods | 1 |
| 12: Control | 0.936 | 1: Electrical machinery, apparatus, and energy | 0.914 | 13: Medical technology | 1 |
| 10: Measurement | 0.903 | 13: Medical technology | 0.838 | 1: Electrical machinery, apparatus, and energy | 0.976 |
| 6: Computer technology | 0.813 | 21: Surface technology and coating | 0.831 | 10: Measurement | 0.976 |
| 2: Audio-visual technology | 0.553 | 2: Audio-visual technology | 0.830 | 21: Surface technology and coating | 0.961 |
| 34: Other consumer goods | 0.370 | 28: Textile and paper machines | 0.785 | 6: Computer technology | 0.949 |
| 4: Digital communication | 0.286 | 6: Computer technology | 0.729 | 29: Other special machines | 0.948 |
| 28: Textile and paper machines | 0.177 | 9: Optics | 0.723 | 2: Audio-visual technology | 0.931 |
| 13: Medical technology | 0.119 | 10: Measurement | 0.722 | 28: Textile and paper machines | 0.927 |
| 21: Surface technology and coating | 0.067 | 29: Other special machines | 0.719 | 23: Mechanical engineering | 0.911 |

According to Table 4, five of the top ten technologies in terms of network eigenvector centrality have EC values of less than 0.5 in the first stage. The EC values of the ten technologies are 0.719 and 0.911. These technologies can be roughly divided into two categories. One is a technology that is always critical to the network, such as 10 (measurement), 6 (computer technology), 2 (audio-visual technology), 34 (other consumer goods), 28 (textile and paper machinery), 13 (medical technology), and 21 (surface technology, coating). These are considered to be critical technologies for smart textile technology. In measurement technologies, such as pressure-sensitive, light-sensitive, and thermal sensors, smart textiles are able to sense changes in their external environment, such as the Lenovo's pressure and temperature regulation smart shoes patent, which collects user movement data and actively collects data that adjusts temperature and pressure. In computer and audio-visual technologies, using cameras and sound collection devices to collect data, in conjunction with computer technology to process the data, renders textiles "intelligent." This includes smart shirts that monitor the wearer's heart rate and respiration. Numerous companies have also conducted research in medical technologies, such as Philips' ECG monitor vest and Healthwatch's abnormal monitoring clothing for pregnant women. The application of various antibacterial, thermal insulation, breathable, waterproof, UV-resistant, and flame-retardant coatings has improved the performance of textiles and spawned new smart textile products such as antibacterial bandages and fireproof clothing. The other category is technology which gradually gains influence due to network evolution. This type of technology has enormous potential for development and deserves consideration in areas such as 1 (motors, instruments, and energy), 29 (other specialized machines), and 23 (chemical engineering). With the advancement of technology, various instruments and types of equipment have become increasingly sophisticated, allowing for the embedding of numerous devices into fabrics and achieving "intelligence." For example, British Broadsword uses conductive fabrics to create power supplies for wearable clothing and Samsung has introduced fabric-based energy generators.

## 5. Discussion and Conclusions

The TC of emerging technologies and traditional industries has developed into a critical innovation mode for a country's competitiveness enhancement. Smart textiles are novel textile products created by combining modern textile technology with chemical, electronic, biological, and other technologies. A smart textile is a classic example of the

fusion of emerging technology and established industrial technology. The network method and life cycle theory were used in this study to investigate the TC of global smart textiles using patent data. First, this study analyzed the life cycle of smart textiles and divided the stages of development into three categories. Then, using the IPC co-occurrence method, it determined the composition of technical fields and constructed the TC network to investigate the structure and key technology of the TC in three distinct stages. The findings indicate that:

(1) After 2021, the TC for smart textiles will reach maturity, and the number of patents will peak in 2030. Similar to the process of life, TC development includes stages such as germination, growth, maturity, and decline. This study used a logistic model to simulate the development stage of smart textile technology and divided the development period into three stages based on the TLC theory.

(2) Using the IPC co-occurrence method, this study identified the fusion technology and analyzed the technical areas covered by smart textiles. Electronic engineering, tools, chemical, mechanical engineering, and others were the primary technical fields involved in smart textiles. Electronic engineering had the highest proportion among them, accounting for 29.11%.

(3) Finally, this study analyzed the structure of the TC based on establishing the TC network of smart textiles. The TC's density, breadth, and depth exhibited upward trends. Measurement, computer technology, and audition technology, for example, have always played significant roles in core technologies, while electrical machinery, instrumentation, energy technology, and other specialized technologies, as well as chemical engineering, have significant development potential.

The above findings have significant implications for the phenomenon of technical convergence that has occurred in the fields of emerging technology and traditional industry. Governments and businesses should promote TC, encouraging research and development organizations to participate in TC to actively shape the future technology paradigm. These findings can aid national policy decisions or company-level strategies when deciding which technologies to promote via TC. Furthermore, the analysis of the technologies' classification structure and convergence patterns can inspire a research funding framework. The findings suggest a policy direction for promoting technological development from a TC standpoint. Moreover, they indicate that the research and funding priorities of research institutions, industries, and academia are concentrated on critical technical fields and that these fields play a role in industrial and technological innovation.

Despite its importance, this study has limitations that may necessitate additional research. First, the data analysis is limited to patent-related data. While patents are the most comprehensive and widely used data sources, additional valuable data sources are available. There is a need to integrate these disparate data sources in the future to obtain valuable information. Second, the method is relatively ineffective at forecasting and can only identify significant fusion technology pairs. In this case, the future integration trend must be bolstered. Finally, applying link forecasting techniques will aid in enhancing the impact of the analysis results on future predictions.

**Author Contributions:** Conceptualization, Q.X. and X.Y.; methodology, Q.X. and Y.Y.; software, Q.X.; validation, Q.X. and Y.Y.; formal analysis, Q.X.; investigation, Q.X. and X.Y; resources, Q.X. and X.Y.; data curation, Q.X.; writing—original draft preparation, Q.X.; writing—review and editing, Q.X. and Y.Y.; visualization, X.Y.; supervision, Y.Y.; project administration, X.Y.; funding acquisition, X.Y. All authors have read and agreed to the published version of the manuscript.

**Funding:** The authors gratefully acknowledge funding from the National Natural Science Fund of China (grant number 71603246) and the National Social Science Foundation of China (21BJY222).

**Institutional Review Board Statement:** Not applicable.

**Informed Consent Statement:** Not applicable.

**Data Availability Statement:** Not applicable.

**Conflicts of Interest:** No potential conflict of interest were reported by the authors.

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
