# Peer review of "Analysis of the Technological Convergence in Smart Textiles"

_sustainability, doi:10.3390/su142013451_

Round 1

Reviewer 1 Report

This is an interesting topic with interesting findings related to the progress in smart textiles and their future. Although the data used for the given statement is too short to justify the findings, still it is important to get these findings.

Author Response

Dear reviewer:

     Thank you very much for your comments and professional advice. Based your suggestion and request, we have made corrected modifications on the revised manuscript. We asked foreign experts in the field to check for grammar and spelling errors. Accordingly, we ensured all the cited references relevant to the research and the article adequately referenced.

  1. This is an interesting topic with interesting findings related to the progress in smart textiles and their future. Although the data used for the given statement is too short to justify the findings, still it is important to get these findings.

The author’s answer: 

Thank you very much for your recognition of our research. We strive to obtain data for a longer period of time in the future research to make the results more reliable.

Yours sincerely

Qian Xu

  1. Oct, 2022

Reviewer 2 Report

1. Generally, the abstract is difficult for the reader; enhance for clarity, including in the abstract, the aim of work, experimental design, some results, and conclusion

2. Type of this article is not clear.

3. Abstract needs to be reformulated to add relevant results, and logistical and structural errors

4. Keywords, add more keywords

5. Rephrase lines 49-70 for clarity

6.  Line 52 rearrange citation according to the date

7. Clear the study objectives at the end of the introduction

8. Line 88 adds a citation.

9. The table needs to be revised and improved.

10. Enhance the resolution of figure 4, 5, and 6

11. Tables lack statical analysis

12. Indicate the abbreviations for the first time

13. All results need to be more presented

14. Check the outputs of all references

15. The style of reference and citation out of the journal structure

There are a lot of typing errors and language mistakes that make me hesitate to accept the publication in its present form.

Author Response

Dear reviewer,

Thank you for giving us the opportunity to modify the paper! Thank you for your helpful suggestions, it is very enlightening for our research. Based on your suggestion and request, we have made corrected modifications on the revised manuscript.

1、Generally, the abstract is difficult for the reader; enhance for clarity, including in the abstract, the aim of work, experimental design, some results, and conclusion

The author’s answer: In this revision, we made improvements in the abstract section. We added the research purpose, research design, results, and conclusions.

2、Type of this article is not clear

The author’s answer: The type of article has been changed.

3、 Abstract needs to be reformulated to add relevant results, and logistical and structural errors

The author’s answer: In this revision, we attempt to make four results of this paper clear.

4、Keywords, add more keywords

The author’s answer: The fifth keyword has been added to the keywords section.

  1. Rephrase lines 49-70 for clarity

The author’s answer: We have revised lines 49-70, as follows:

  Smart textiles are woven, spun, or braided using smart fibres or other smart materials [11,12,13] . As the key application object of the industrial Internet as well as the excellent carrier of intelligent services, smart textiles have generated considerable interest in the industry. The United States Joint Market Research (AMR) reveals that, the global smart textile market will grow from US $943 million in 2015 to US $5.369 billion in 2022.

This study aims to investigate the fusion between textile technology and emerging digital technology. Specifically, what is the technological life cycle of smart textiles? What technical fields are involved in smart textiles’ TC? What is the structure of the TC? What are the primary technologies? For the purpose of achieving these objectives, the life cycle of smart textiles is classified into three development stages. The composition, structure, and key technologies of TC in each stage are then determined by utilising the social network method. The significance of our research is: (1) to lay a theoretical foundation for the future technology integration of emerging technologies and traditional industries. (2) Network analysis indicators can be employed to establish the relationship between technical fields and determine the rules of TC while providing academic reference. (3) By analysing the technological cooperation trend of emerging technologies in the textile industry, new technological opportunities can be found, and future R&D trends can be predicted, which will help the government in formulating industrial policies.

  1. Line 52 rearrange citation according to the date

The author’s answer:  We have revised accordingly

 (Cherenack&Pieterson, 2012; Chan et al., 2012 ; P et al., 2021;)

  1. Clear the study objectives at the end of the introduction

The author’s answer:  we have added study objectives at the end of the introduction.

This study aims to investigate the fusion between textile technology and emerging digital technology. Specifically, what is the technological life cycle of smart textiles? What technical fields are involved in smart textiles’ TC? What is the structure of the TC? What are the primary technologies?

  1. Line 88 adds a citation.

The author’s answer: The Line88 references that were missing were added.

  1. The table needs to be revised and improved.

The author’s answer: We have carefully revised each table in the article in order to make the contents of the table clearer.

  1. Enhance the resolution of figure 4, 5, and 6

The author’s answer: We have verified that the resolution of Figures 4, 5, and 6 has been adjusted. For detail, please refer to the revised article.

  1. Tables lack statical analysis

The author’s answer:We tried our best to analyze the details in the table.

  1. Indicate the abbreviations for the first time

The author’s answer: We carefully checked the first reference of TC, TLC, and other abbreviations, and tried our best to mention the abbreviations the first time.

  1. All results need to be more presented

The author’s answer: We place great importance on your suggestions, thus revised the section to make the results more detailed and understandable. Please refer to the result section of the revised manuscript.

  1. Check the outputs of all references

The author’s answer: Each cited document was reviewed to ensure correct output.                               

  1. The style of reference and citation out of the journal structure

The author’s answer: We modified the citation format to match the style of this journal.

16.

In addition, we asked foreign experts in the field to check for grammar and spelling errors.

Reviewer 3 Report

Reference 28 is not existing

check the spelling of Degree 

Author Response

Dear reviewer:

     Thank you very much for your recognition of our research .Thank you very much for your comments and professional advice. Based your suggestion and request, we have made corrected modifications on the revised manuscript. We asked foreign experts in the field to check for grammar and spelling errors.

  1. Reference 28 is not existing

The author’s answer: We rechecked the citation, and now each cited document in our paper was reviewed to ensure correct output.

  1. Check the spelling of Degree .

The author’s answer:We checked the spelling of degree to make sure it was correct.

Yours sincerely

Qian Xu

  1. Oct, 2022

Round 2

Reviewer 2 Report

The authors have carefully processed all comments. The quality of the manuscript has increased significantly. I have no further comments.